

# Organic signatures in Pleistocene cherts from Lake Magadi (Kenya), analogs for early Earth hydrothermal deposits

Manuel Reinhardt[1,2], Walter Goetz[1], Jan-Peter Duda[3], Christine Heim[2], Joachim Reitner[2,4], Volker Thiel[2]

[1]Planets and Comets, Max Planck Institute for Solar System Research, 37077 Göttingen, Germany
[2]Department of Geobiology, Geoscience Centre, University of Göttingen, 37077 Göttingen, Germany
[3]Department of Earth Sciences, University of California Riverside, CA 92521, USA
[4]'Origin of Life' Group, Göttingen Academy of Sciences and Humanities, 37073 Göttingen, Germany

*Correspondence to*: Manuel Reinhardt (mreinha@gwdg.de)

**Abstract.** Organic matter in Archean hydrothermal cherts may provide an important archive for molecular traces of earliest life on Earth. The geobiological interpretation of this archive, however, requires a sound understanding of organic matter preservation and alteration in hydrothermal systems. Here we report on organic matter (including molecular biosignatures) enclosed in hydrothermally influenced cherts of the Pleistocene Lake Magadi (Kenya; High Magadi Beds and Green Beds)—important analogs for Archean cherts. The Magadi cherts contain low organic carbon (<0.4 wt.%) that occurs in form of finely dispersed clots, layers, or encapsulated within microscopic carbonate rhombs. Both, extractable (bitumen) and non-extractable organic matter (kerogen) was analyzed. The bitumens contain immature "biolipids" like glycerol mono- and diethers (e.g., archaeol and extended archaeol), fatty acids and –alcohols indicative for, *inter alia*, thermophilic cyanobacteria, sulfate reducers, and haloarchaea. However, co-occurring "geolipids" such as *n*-alkanes, hopanes, and polycyclic aromatic hydrocarbons (PAHs) indicate that a fraction of the bitumen has been thermally altered to early or peak oil window maturity. This more mature fraction likely originated from defunctionalization of dissolved organic matter and/or hydrothermal petroleum formation at places of higher thermal flux. Like the bitumens, the kerogens also show variations in thermal maturities, which can partly be explained by admixture of thermally pre-altered macromolecules. However, findings of archaea-derived isoprenoid moieties in some of the kerogens indicate that a fast sequestration of microbial lipids into kerogen must have occurred while hydrothermal alteration was active. We posit that such early sequestration may enhance the survival of molecular biosignatures during *in-situ* hydrothermal (and post-depositional) alteration through deep time. Furthermore, the co-occurrence of organic matter with different thermal maturities in the Lake Magadi cherts suggests that similar findings in Archean hydrothermal deposits could partly reflect original environmental conditions, and not exclusively post-depositional overprint or contamination. Our results support the view that kerogen in Archean hydrothermal cherts may contain important information on early life. Our study also highlights the suitability of Lake Magadi as an analog system for hydrothermal chert environments on the Archean Earth.

# 1 Introduction



Organic matter trapped in Archean cherts is of utmost relevance for the reconstruction of earliest microbial processes on Earth, but its origin is only poorly constrained. Diagenesis and metamorphic processes have been obliterating the original organic matter over billions of years and complicate its interpretation. Many of the Archean cherts are associated with hydrothermal settings (e.g., Brasier et al., 2002; Djokic et al., 2017; Duda et al., 2016, 2018; Hickman-Lewis et al., 2018). In

such environments, organic compounds may rapidly decompose due to elevated temperature and pressure conditions (Hawkes et al., 2015, 2016; Rossel et al., 2017) and may also be redistributed via hydrothermal cycling in the form of bitumen (e.g., Weston and Woolhouse, 1987; Clifton et al., 1990; Leif and Simoneit, 1995) or kerogen (Duda et al., 2018). The interpretation of organic signatures in Archean hydrothermal cherts therefore requires detailed knowledge on the preservation, alteration, and distribution pathways of organic matter in such environments. Some of these aspects can be

studied in modern analogs.

Archean cherts generally originate from chemical precipitation or replacement processes of silica rather than biogenic precipitation by silicifying organisms (e.g., Sugitani et al., 2002; van den Boorn et al., 2007). Siliceous sediments associated with chemical precipitation are rare on the modern Earth, but can be found in some hot spring or shallow lacustrine environments. Important sites include the Taupo Volcanic Zone (New Zealand; Campbell et al., 2003), the Geysir hot spring

area (Iceland; Jones et al., 2007; Jones and Renaut, 2010), the El Tatio geothermal field (Chile; Jones and Renaut, 1997; Nicolau et al., 2014) and the East African Rift system (Kenya; Renault et al, 2002). Among the latter, the alkaline chert environment of Lake Magadi is of particular interest, as it represents an analog for Archean hydrothermal chert environments (Eugster and Jones, 1968; Pirajno and Van Kranendonk, 2005; Brenna, 2016).

Lake Magadi, the focus of this study, is located in the lowermost depression of the East African Rift Valley (south Kenya;

ca. 1°54' S, 36°16' E). The geology of the surrounding hills is dominated by alkali trachyte (1.65 to 0.8 Ma; Baker, 1958, 1986). Cherts occur in three sedimentary units overlaying the trachytes, namely the Oloronga Beds (ca. 0.8 to 0.3 Ma; Fairhead et al., 1972; Behr and Röhricht, 2000), the Green Beds (*sensu* Behr and Röhricht 2000; Behr 2002; ca. 40 to 100 ka; Goetz and Hillaire-Marcel, 1992; Williamson et al., 1993) and the High Magadi Beds (9 to 25 ka; Williamson et al., 1993; Tichy and Seegers, 1999). Each of these units represents a different lake stage. Today, trona ($Na_3(HCO_3)(CO_3 \cdot H_2O)$) is

precipitating in large areas of the residual lake (Evaporite Series; Baker, 1958), and Lake Magadi is the type locality for cherts based on the sodium silicate mineral magadiite ($NaSi_7O_{13}(OH)_3 \cdot 4(H_2O)$); Eugster, 1967, 1969; Eugster and Jones, 1968; Hay, 1968).

Lake Magadi has strongly been influenced by changes in the local climate. Today the Magadi basin represents an evaporation pan with a closed hydrological cycle (i.e., no outflow) that is only recharged by ephemeral runoff and

hydrothermal springs (ca. 28 to 86 °C at present; Eugster, 1970; Jones et al., 1977). During the Pleistocene, however, the water level has changed several times. The Oloronga Beds (not investigated in this study) were formed in a stratified freshwater lake (Roberts et al., 1993). The Green Beds, in contrast, were deposited in a highly dynamic, alkaline, shallow water environment, probably in periodically flooded hot spring mudflats (Behr and Röhricht, 2000). Lake levels may then have increased again during deposition of the High Magadi Beds (Behr and Röhricht, 2000), but the setting remained



strongly evaporitic. However, chemical precipitation of silica gels, the precursor of most cherts at Lake Magadi, was likely induced by hydrothermal processes (Eugster and Jones, 1968), evaporation and microbial activity (Behr and Röhricht, 2000; Behr, 2002).

A variety of cherts from Lake Magadi and its surroundings contain microbial structures (Behr and Röhricht, 2000; Behr, 2002; Brenna, 2016). Especially the Green Bed cherts are associated with fingerprints of microbial activity such as stromatolites and silicified cyanobacteria cells (Behr and Röhricht, 2000). Organic matter archived in these cherts potentially encodes important information of geobiological value but has not been characterized so far.

Our study is focused on the origin, alteration and preservation of the organic matter in Pleistocene hydrothermal cherts from Lake Magadi, to support the interpretation of organic matter in early Earth hydrothermal deposits. For our analyses we used several complementary petrographic and organic-geochemical techniques, including (scanning electron) microscopy, Raman spectroscopy, catalytic hydropyrolysis (HyPy), gas chromatography-mass spectrometry (GC-MS) and gas chromatography-combustion-isotope ratio mass spectrometry (GC-C-IRMS). The combined application of petrographic and geochemical techniques is required to fully understand organic matter characteristics (e.g., appearance on macroscopic and molecular levels, identification of heterogeneities) in context of the depositional environment.

## 2 Materials and Methods

### 2.1 Sample material, petrographic and bulk geochemical analyses

Cherts from the Pleistocene High Magadi Beds (LM-1692–1695) and Green Beds (LM-1696–1699) were sampled from different surficial outcrops around the present Lake Magadi (extent area of the Pleistocene Lake Magadi; Röhricht, 1999). A recent siliceous sinter from Great Geysir, Iceland (IC-1700; 64°18'46'' N, 20°18'03'' W) was additionally analyzed as a reference.

Petrographic observation was performed on thin sections using a Zeiss SteREO Discovery.V8 stereomicroscope connected to an AxioCam MRc5 5-megapixel camera (transmitted and reflected light) and with a Leica DMLP microscope coupled to a Kappa Zelos-655C camera (polarized light). Chert fragments (sputtered with Au-Pd, 7.3 nm for 120 s) were furthermore investigated using a LEO 1530 Gemini scanning electron microscope (SEM) coupled with an Oxford INCA X-act energy dispersive X-ray spectrometer (EDX). Contents of organic carbon ($C_{org}$), inorganic carbon ($C_{inorg}$), sulfur and nitrogen were determined with a Hekatech Euro EA elemental analyzer and a Leco RC612 temperature programmable carbon analyzer. Element distributions were analyzed on sample slices using a Bruker M4 Tornado µ-XRF scanner equipped with a rhodium target X-ray tube at 50 kV and 200 µA. Areas of ca. 50 mm$^2$ were mapped with scan resolution of 500x300 and using a spot size of 20 µm.

### 2.2 Organic-geochemical preparation



All materials used for biomarker preparation were heated to 500 °C (3 hrs) and/or carefully rinsed with acetone. A blank (pre-combusted sea sand) was processed in parallel to track potential laboratory contamination. Outer surfaces (2–5 mm) of the chert samples were removed with a particularly cleaned rock saw, and the surfaces of the resulting inner blocks were carefully rinsed with dichloromethane (DCM). After grinding (Retsch MM 301 pebble mill), sample powders (50 g each) were ultrasonically extracted with 100 mL DCM/methanol (MeOH) (2/1, v/v), 100 mL DCM/MeOH (3/1, v/v) and 100 mL DCM (10 min, respectively). The total organic extract (TOE) was then desulfurized with reduced Cu. 10 % of each TOE was derivatized with trimethylchlorosilane (TMCS)/MeOH (1/9, v/v; heated for 1 h 30 min at 80 °C) and subsequently with $N,O$-bis(trimethylsilyl)trifluoroacetamide (BSTFA)/pyridine (3/2, v/v; heated for 1 h at 40 °C) to convert carboxyl groups into methyl esters, and hydroxyl groups into trimethylsilyl ethers. Another 50 % of each TOE was separated via column chromatography into a hydrocarbon (F1), alcohol/ketone (F2; including free lipids) and a polar fraction (F3). In brief, 7 g silica gel 60 were filled into a glass column (1.5 cm internal diameter), plugged with pre-extracted cotton wool and sand. The dried TOE was vapor-deposited onto ca. 0.5 g silica gel 60 and added to the column. F1 was eluted with 20 mL $n$-hexane/DCM (8/2, v/v), F2 with 30 mL DCM/ethyl acetate (9/1, v/v) and F3 with 100 mL DCM/MeOH (1/1, v/v) and 100 mL MeOH. F2 and F3 were derivatized with TMCS/MeOH (1/9, v/v; heated for 1 h 30 min at 80 °C) and with BSTFA/pyridine (3/2, v/v; heated for 1 h at 40 °C).

The extraction residues were decalcified with HCl (37 %, 1 d, 20°C) and desilicified with HF (48 %, 7 d, 20°C). The remaining kerogens (i.e., the non-extractable portion of organic matter: Durand, 1980) from the Magadi cherts were used for catalytic hydropyrolysis (HyPy; Sec. 2.3; LM-1692–1693, LM-1695, LM-1697–1698) and Raman spectroscopy (Sec. 2.6; LM-1692–1699). No kerogen could be isolated from IC-1700.

## 2.3 Catalytic hydropyrolysis (HyPy)

HyPy is an open-system pyrolysis technique for studying the molecular kerogen composition (Love et al., 1995). It involves the gentle release of kerogen-bound compounds through progressive heating under a high-pressure hydrogen atmosphere (150 bar) and in the presence of a sulfided molybdenum catalyst (ammonium dioxydithiomolybdate). HyPy has been demonstrated to be very sensitive and leaving the organic stereochemistry of released compounds largely intact (e.g., Love et al., 1995, 1997; Bishop et al., 1998; Meredith et al., 2014).

Our experiments were conducted with a HyPy device from Strata Technology Ltd. (Nottingham, UK), following existing protocols (e.g., Brocks et al., 2003b; Marshall et al., 2007; Duda et al., 2018). In brief, between 1−10 mg of pre-extracted kerogen (3x ultrasonically extracted in DCM/MeOH (3/1, v/v)) was loaded with 10 wt.% of the catalyst and then pyrolyzed under a constant hydrogen flow of 6 L min$^{-1}$. HyPy was conducted following a two-step approach. The first step involved heating of the kerogens from ambient temperature to 250 °C (at 300 °C min$^{-1}$) and then to 330 °C (at 8 °C min$^{-1}$, held for 10 min). During this step, residual bitumens and compounds bound via unstable covalent bonds were released. In a second step, the remaining kerogens were heated from ambient temperature to 520 °C (at 8 °C min$^{-1}$), releasing solely covalently bound molecules. Pyrolysates from all steps were collected in a silica gel trap cooled with dry ice (Meredith et al., 2004) and



subsequently analyzed via GC-MS (Sec. 2.4) and GC-C-IRMS (Sec. 2.5). Blanks were run regularly to ensure constant experimental conditions and track potential contamination.

## 2.4 Gas chromatography-mass spectrometry (GC-MS)

Molecular fractions were analyzed using a Thermo Trace 1310 gas chromatograph (GC) coupled to a Thermo TSQ Quantum
Ultra triple quadrupole mass spectrometer (MS). The GC was equipped with a fused silica capillary column (Phenomenex Zebron ZB-5MS, 30 m length, 250 μm internal diameter, 0.25 μm film thickness). Samples were injected with a Thermo TriPlus RSH autosampler into a splitless injector and transferred to the GC column at 320 °C. The GC oven was heated under a constant He flow (1.5 mL min$^{-1}$) from 80 °C (held for 1 min) to 325 °C at 5 °C min$^{-1}$ (held for 30 min). The MS source operated in electron ionization mode at 70 eV and 240 °C. Organic compounds were analyzed in full scan mode (scan
range 50–850 amu) and identified by comparison with published retention times and mass spectra.

## 2.5 Gas chromatography-combustion-isotope ratio mass spectrometry (GC-C-IRMS)

Compound-specific stable carbon isotope ratios ($\delta^{13}C_{V-PDB}$) were measured using a Thermo Scientific Trace GC coupled to a Delta Plus isotope ratio mass spectrometer (IRMS) via a combustion reactor (C). The GC was equipped with two serially linked silica capillary columns (Agilent DB-5 and DB-1, each with 30 m length, 250 μm internal diameter, 0.25 μm film
thickness). The combustion reactor contained CuO, Ni and Pt, and was operated at 940 °C. Fractions were injected into a splitless injector and transferred to the GC column at 290 °C. The carrier gas was He with a flow rate of 1.2 mL min$^{-1}$. The temperature program started at 80 °C, followed by heating to 325 °C at 5 °C min$^{-1}$ (held for 60 min). Laboratory standards were analyzed to control the reproducibility of measuring conditions and $CO_2$ gas of known isotopic composition was used for calibration. $\delta^{13}C$ of the MeOH used for derivatization (methyl esthers) and androstanol (underivatized, as well as TMS
derivative) were measured to track isotopic changes through derivatization. The $\delta^{13}C$ values of derivatized compounds were then corrected according to Goñi and Eglinton (1996).

## 2.6 Raman spectroscopy

Raman spectroscopy was conducted on sample slices (thickness ca. 1 cm) and isolated kerogen flakes (see Sec. 2.2). At least 10 measurements were conducted per sample in order to evaluate internal variation and identify potential outliers. The
measurements were performed with a WITEC alpha300 instrument. Spectra (scan range 100–4000 rel. cm$^{-1}$) were generated with a frequency-doubled continuous-wave Nd-YAG laser (532 nm, beam intensity of ca. 1 mW) over 10 s integration time by focusing through a 50x optical power objective. The reflected beam was dispersed by a 600 L mm$^{-1}$ grating on its way into the detector (CCD, 248 pixels). The diameter of each measurement spot was ca. 0.7 μm. Calibration of the instrument (including stability over time) was routinely controlled by pure mineral phases (quartz, calcite, gypsum, beryl). The WITEC
Control and Project FOUR 4.1 software was used to record and process all spectral data (smoothing, baseline correction, fitting). Choosing an appropriate Raman fitting method for organic carbon is not trivial as the established methods refer to





specific temperature windows. In this study we applied a 6-Voigt-functions fit (after Schito et al., 2017) for low-temperature organic carbon and a 4-Voigt-functions fit (Beyssac et al., 2002) for high-temperature organic carbon. Band nomenclature for the low-temperature organic carbon follows Rebelo et al. (2016), while the high-temperature organic carbon bands are named after Beyssac et al. (2002). Peak temperatures ($T_{max}$; averaged over all valid measurements on each sample) were

5 inferred from (i) the vitrinite reflectance $R_0$ that in sum was calculated from the Raman band ratio RA2 for low-thermal-maturity spectra (RA2=(S+Dl+D)/(Dr+Gl+G); Schito et al., 2017; Barker and Pavlewicz, 1994) and (ii) the Raman band ratio R2 for high-thermal-maturity spectra (R2=D1/(G+D1+D2); Beyssac et al., 2002).

## 3 Results

### 3.1 Petrography, bulk-geochemistry and Raman spectroscopy

Most of the Lake Magadi cherts studied reveal a dense silica matrix, except LM-1692 and LM-1693 which show microscopic pores of <50 µm (Fig. 1a, d). The cherts exhibit brecciated (Fig. 1f), cloudy (Fig. 1g), and laminated (Fig. 1i–k) textures. Most of the textures resemble microbial mat fabrics and can contain distinct silicified microbial cells and filaments (Fig. 1e).

The samples show $C_{org}$ values between 0.01 and 0.34 wt.% and CaCO$_3$-contents of 0.05 to 4.47 wt% (Table 1). Total
15 nitrogen (N) and sulfur (S) contents are generally low (<0.02–0.05 wt.%, respectively; Table 1).

The organic matter occurs either layered (up to 0.5 mm; Fig. 1a, h), or finely dispersed in the form of small clots in the chert matrix (<20 µm; Fig. 1b–c). In some samples, organic matter is also associated with carbonate aggregates (Fig. 1m–q) and sulfur enrichments (Fig. 1q). The carbonate aggregates are up to 1 mm in size and partly have a rhombic shape (e.g., in LM-1696: Fig. 1m–q).

Raman spectra of isolated kerogen particles show a broad D-band centered at ca. 1354 cm$^{-1}$ and a G-band at ca. 1597 cm$^{-1}$ (Fig. 2). Vitrinite reflectance $R_0$ (calculated from RA2; Schito et al., 2017) generally ranges between 0.32 and 0.72 %, corresponding to maximum temperatures in the range of 40–110 °C (Table 2). Sample LM-1697 exhibited a second kerogen population with D- and G-bands centered at 1357 and 1577 cm$^{-1}$, respectively (Fig. 2), corresponding to a maximum temperature of ca. 440 °C (high-temperature, graphitic; Table 2, Fig. 2c; Beyssac et al., 2002).

### 3.2 Bitumen

Figure 3 shows GC-MS chromatograms from bitumens of High Magadi Bed cherts (Fig. 3a–c), Green Bed cherts (Fig. 3d, e), and a recent siliceous sinter from Great Geysir in Iceland (Fig. 3f). The most noticeable compound classes in all samples are *n*-alkanes, *n*-alkanoic acids and *n*-alkan-1-ols (in decreasing abundance), plus glycerol diethers (Fig. 3a–c). GC-amenable aromatic compounds are low in abundance, but some polycyclic aromatic hydrocarbons (PAHs) were identified in all
30 samples. One sample showed a pronounced unresolved complex mixture (UCM; LM-1697; Fig. 3d).



### 3.2.1 Functionalized lipids

*Fatty acids (alkanoic and alkenoic acids)*

$n$-Alkanoic acids typically range from $C_{12}$ to $C_{32}$ (Fig. 4a) and exhibit a clear even-over-odd-predominance, as expressed by OEP values <<1 (odd-to-even-predominance; Scalan and Smith, 1970; Table 3). The most abundant fatty acids are $n$-hexadecanoic acid ($C_{16:0}$) and $n$-octadecanoic acid ($C_{18:0}$). $n$-Alkenoic acids occur at low abundance and include $C_{16:1}$, $C_{18:1}$ (tentatively identified as ω9c/9t) and $C_{18:2}$. Terminally methylated (*iso*($i$)- and *anteiso*($ai$)-) alkanoic acids are also present, including $i$-$C_{15:0}$ and $i$-$C_{17:0}$ (LM-1692–1694) plus $ai$-$C_{14:0–17:0}$ and $ai$-$C_{24:0–25:0}$ (LM-1692–1694). IC-1700 additionally shows $i$-$C_{16:0}$. Phytanic acid occurs in LM-1692–1695. The $\delta^{13}$C signatures of the short- ($C_{12–18}$) and long-chain ($C_{24–28}$) alkanoic acids range from −22.4 to −29.6 ‰ in the Magadi cherts, and from −25.2 to −32.5 ‰ in IC-1700 (Table 4). The values do not differ much between short- and long-chain homologues ($\Delta$ < 2.4 ‰), except for IC-1700 ($\Delta$ = 7.3 ‰).

*Alkanols and alkanones*

$n$-Alkan-1-ols typically range from $C_{12}$ to $C_{32}$ (Fig. 4b). These compounds show a strong even-over-odd-predominance in all samples (OEP17 between 0.1 and 0.2, OEP29 between <0.1 and 0.3; Table 3). Hexadecan-1-ol ($C_{16}$-OH) and octadecan-1-ol ($C_{18}$-OH) show highest abundances. In addition to the $n$-homologues, odd-numbered $C_{15–25}$ $ai$-alkan-1-ols are present (see Fig. 4b). Short-chain ($C_{12}$ to $C_{18}$) $n$-alkan-1-ols from the Magadi cherts reveal mean $\delta^{13}$C values between −29.4 and −35.9 ‰ (−27.7 ‰ in IC-1700). With increasing chain length, the homologues get more enriched in $^{13}$C ($C_{24–28}$, $\Delta$ up to 15.7 ‰ in LM-1694; Table 4).

$n$-Alkan-2-ols occur in diverse ranges (e.g., $C_{15}$ to $C_{31}$ in LM-1698; Fig. 5b). The distributions are unimodal (maximum at $C_{20}$) in LM-1692–1696, or bimodal (maxima at $C_{20}$ and $C_{31}$) in LM-1697–1699. In all samples, medium-chain $n$-alkan-2-ols ($C_{18–24}$) have no chain-length-predominance (OEP21 between 0.8 and 1.1; Table 3), while long-chain homologues ($C_{25–31}$) exhibit a clear odd-over-even-predominance (OEP29 between 1.9 and 5.9). IC-1700 shows a bimodal distribution with maxima at $C_{14}$ (minor) and $C_{20}$ , and no chain-length-predominance.

$n$-Alkan-2-ones typically appear in the range of $C_{15}$ to $C_{31}$, but are virtually absent in LM-1699. Some samples show a unimodal distribution with no chain-length-preference (Fig. 5; Table 3), while most reveal a slight odd-over-even-predominance (OEP21 between 1.3 and 2.1). The isoprenoid ketone 6,10,14-trimethyl pentadecan-2-one additionally occurs in every sample and is the most abundant alkan-2-one in LM-1692–1695 and IC-1700. Furthermore, $i$- and $ai$-alkan-2-ones appear in LM-1698 ($C_{18}$ to $C_{23}$; Fig. 5c).

*Other lipids*

Glycerol monoethers (1-*O*-alkylglycerols) occur in all samples from Lake Magadi. Their highest diversity is observed in LM-1692–1694, including methyl-branched ($i$-$C_{16:0}$, 10Me-$C_{16:0}$, $i$-$C_{17:0}$, $ai$-$C_{17:0}$, Me-$C_{17:0}$, $i$-$C_{18:0}$), and straight ($C_{15–18}$) alkyl chains (see Fig. 4c). The most prominent monoether is 1-*O*-(10-methyl)-hexadecylglycerol (10Me-$C_{16:0}$). Furthermore, two glycerol diethers, namely di-*O*-phythanylglycerol (archaeol; "A", Fig. 3a–c) and *O*-phytanyl-*O*-sesterterpanylglycerol





(extended archaeol; "ExA", Fig. 3a–c) appear in LM-1692–1696 and LM-1699. Mono- and diethers show $\delta^{13}C$ values between −10.9 and −22.2 ‰ (Table 4; Fig. 3a–c), with highest values in LM-1694 (−10.9 and −12.2 ‰, respectively). Additionally, functionalized sesqui- and diterpenoids are always present and traces of $C_{31}$ or $C_{32}$ hopanoic acids are found in some samples. LM-1693 and LM-1695–1699 furthermore contain abundant tetrahymanol ($\delta^{13}C$ between −24.1 and −33.3

5    ‰). Sterols, particularly cholesterol and sitosterol, appear in small amounts in most samples.

### 3.2.2 Aliphatic hydrocarbons

*n*-Alkanes range from *n*-$C_{15}$ to *n*-$C_{33}$ and primarily show a unimodal distribution (maximum around *n*-$C_{21}$ and *n*-$C_{22}$; Figs. 3, 5a, S1) and no carbon chain-length-preference up to *n*-$C_{25}$ (OEP21 between 1.0 and 1.2; Table 3). However, an odd-over-even-preference is always observed for greater chain-lengths (OEP31 between 1.8 and 7.0; Table 3). Furthermore, *i*- and *ai*-

alkanes are present ($C_{18}$ to $C_{25}$; Fig. 5a), following the distribution trend of the corresponding *n*-alkanes. Pristane (Pr) and phytane (Ph) are visible in all samples except LM-1699 (only Ph; Fig. S1). Pr/Ph ratios are below 0.37, while Ph/*n*-$C_{18}$ ratios range between 0.26 and 0.49 (Table 2). LM-1696 furthermore reveals 6-methylheptadecane (6Me-$C_{17}$; Fig. S1e). Medium-chain *n*-alkanes ($C_{17–24}$) show mean $\delta^{13}C$ values between −29.7 and −33.3 ‰ in the Lake Magadi cherts (−35.7 ‰ in IC-1700), while $\delta^{13}C$ values of higher homologues (>$C_{24}$) increase up to −26.2 ‰ (Δ between 1.9 and 7.2 ‰; Table 4).

The samples furthermore contain traces of 17α,21β-hopanes (S+R isomers). The S/S+R isomer ratios of the $C_{31}$ pseudohomologues range between 0.49 and 0.61 (Table 2). Steranes are below detection limit.

### 3.2.3 Polycyclic aromatic hydrocarbons (PAHs)

All samples contain low amounts of (monomethyl-) phenanthrenes, while anthracene is only observed in IC-1700. The methylphenanthrene indices (MPI-1, after Radke and Welte, 1983) vary between 0.48 and 1.02, resulting in calculated

vitrinite reflectances ($R_c$, after Boreham et al., 1988) between 0.56 and 0.94 % (Table 2). Traces of dimethylphenanthrenes are detected in LM-1692–1698. Other PAHs observed are fluoranthene (Flu) and pyrene (Py) with Flu/(Flu+Py) ratios between 0.48 and 0.96 (Table 2).

### 3.3 Kerogen (high temperature HyPy step, up to 520 °C)

The high temperature HyPy pyrolysates (up to 520 °C; see Sec. 2.3) can be divided in two groups according to their

compositions. LM-1692 and LM-1693 show a strong aromatic character (aliphatics/aromatics of 0.4 and 0.2, respectively), which is not observed in LM-1695 and LM-1697–1698 (aliphatic/aromatic of 1.1, 1.0 and 1.5, respectively). All pyrolyzed kerogens reveal varying distributions of *n*-alkanes (Fig. 6; see below).

### 3.3.1 Aliphatic hydrocarbons

Kerogen-bound *n*-alkanes exhibit maxima around *n*-$C_{18}$ (LM-1693, LM-1695, LM-1698), *n*-$C_{21}$ (LM-1692–1693, 1697) and

*n*-$C_{32}$ (LM-1695, LM-1697–1698), and range from *n*-$C_{18}$ to *n*-$C_{36}$ (LM-1692–1693), *n*-$C_{14}$ to *n*-$C_{44}$ (LM-1695) or *n*-$C_{16}$ to *n*-



$C_{46}$ (LM-1697–1698). No carbon chain-length-preference is visible up to $n$-$C_{26}$ (OEP21 always 1.0, except for LM-1695; Table 3; Fig. 6c), but a slight even-over-odd-preference is observed for longer chains (OEP31 between 0.7 and 0.9; Table 3). Moreover, all pyrolysates contain few $i$- and $ai$-alkanes (Fig. 6). Mean $\delta^{13}C$ values of medium-chain $n$-alkane moieties ($C_{17–24}$) range from −23.5 to −34.2 ‰, whereas long-chain $n$-alkanes ($C_{25–40}$) reveal values between −21.9 and −30.2 ‰ (Δ between 1.3 and 7.1 ‰; Table 4).

The regular acyclic isoprenoids phytane (Ph) and 2,6,10,14,18-pentamethylicosane (PMI$_{reg}$; identified via mass spectrum; Fig. S2; Risatti et al., 1984; Greenwood and Summons, 2003) appear in LM-1692–1693 and LM-1695 (Ph/$n$-$C_{18}$ between 0.49 and 1.89). The regular acyclic isoprenoids Farnesane (Far), norpristane (Nor) and pristane (Pr) are only present in LM-1695 (Pr/Ph = 0.24; Table 2), and biphytane occurs in LM-1693 and LM-1695 (Fig. 6b–c). The detection of phytane and PMI$_{reg}$ in the kerogens of LM-1692–1693 and LM-1695 coincides with the appearance of archaeol and extended archaeol in the corresponding bitumens (Fig. 3a–c). $\delta^{13}C$ values of PMI$_{reg}$ vary between −22.0 and −24.6 ‰, while phytane exhibts $\delta^{13}C$ values between −25.1 and −28.5 ‰ (Table 4; Fig. 6a–c).

### 3.3.2 PAHs

All kerogen pyrolysates contain (mono- and dimethylated) phenanthrenes, anthracene, plus various 4- and 5-ring PAHs. MPI-1 ranges from 0.89 to 1.69, corresponding to $R_c$ values between 0.85 and 1.41 % (Table 2). Fluoranthene (Flu) and pyrene (Py) with Flu/(Flu+Py) ratios between 0.23 and 0.44 are also present. Methyl naphthalenes only occur in LM-1693 and LM-1695, while di- and trimethyl naphthalenes appear in LM-1693, LM-1695 and LM-1698.

## 4 Discussion

### 4.1 Thermal maturity and syngeneity of the organic matter

The studied Lake Magadi cherts are of Pleistocene age and have not been buried. This is in good accordance with several molecular characteristics of the bitumens that suggest an immature nature of the organic matter. These features include the OEP29 of $n$-alkanoic acids (0.3–0.5) and $n$-alkan-1-ols (<0.1–0.3), the OEP31 of $n$-alkanes (2.3–7.0), and the presence of intact functionalized lipids (e.g., archaeol, extended archaeol and monoethers). On the other hand, the OEP21 of medium-chain $n$-alkanes (1.0–1.2), Ph/$n$-$C_{18}$ ratios (≤0.49), MPI-1 ratios (0.48–1.02, mean 0.75), $R_c$ values (0.56–0.94 %, mean 0.74 %) and $C_{31}$ S/(S+R) ratios (0.49–0.61, mean 0.56) are in line with early to peak oil window maturity (see ten Haven et al., 1987; Killops and Killops, 2005; Peters et al., 2005; Table 2). Hence, the bitumen preserved in the Magadi cherts consists of at least two fractions, a "fresh" immature portion co-occurring with a thermally mature component.

A similar maturity offset is also reflected in bulk- and molecular kerogen characteristics. A low thermal maturity is for instance indicated by low Raman-derived $T_{max}$-signatures in some samples (LM-1697 and LM-1698; ca. 40 and 50 °C; Fig. 2c; Table 2), and a slight even-over-odd preference of long-chain $n$-alkanes in all kerogen pyrolysates (OEP31 between 0.7 and 0.9; Table 3). At the same time, MPI-1 ratios (≤1.69), $R_c$ values (≤1.41 %), the OEP21 of medium-chain $n$-alkanes (1.0





= no preference) indicate an elevated thermal maturity, which is in good accordance with Raman temperatures from the High Magadi Bed cherts and LM-1696 ($T_{max}$ of up to 110 °C; Table 2). Some of the Raman spectra from the LM-1697 kerogen even evidence the presence of a high-temperature graphitic component (i.e., $T_{max}$ ~440 °C; Fig. 2d, Table 2).

Such offsets between different thermal maturity parameters are typically related to an emplacement of organic material from another source (e.g., modern endoliths; e.g., Golubic et al., 1981; Hallmann et al., 2015). Most of the Lake Magadi cherts studied reveal a dense silica matrix, but a few samples (LM-1692, LM-1693) indeed show small pores that would allow for such emplacements. However, a recent emplacement is unlikely for the following reasons:

(i)     The analyzed samples did not show any viable microbial colonization (e.g., biofilms, or endolith borings).

(ii)    No carbonaceous microbial remains were discovered via SEM coupled to EDX and all detected microfossils are silicified (see Fig. 1e).

(iii)   Kerogens contain fingerprints of functionalized moieties in their corresponding bitumens (e.g., isoprenoids appear only in kerogens that show archaeol and extended archaeol in their corresponding bitumens; Figs. 3 and 6).

(iv)    The $\delta^{13}C$ values of long-chain $n$-alkanes from the Green Bed chert kerogens matches the $\delta^{13}C$ values of long-chain $n$-alkanoic acids from bitumens ($\Delta \leq 3$ ‰), supporting their taphonomic relation.

Consequently, both, the rather immature and the thermally altered organic matter can be considered syngenetic to the Pleistocene cherts.

## 4.2 Geobiology of the Lake Magadi during chert deposition

### 4.2.1 Prokaryotes

Archaeol and extended archaeol appear in all High Magadi Bed and two Green Bed chert bitumens (LM-1696 and LM-1699), and their molecular fossils are important contributors to the corresponding kerogens. While archaeol is a common constituent of Euryarchaeal lipids (e.g., Koga, 1993; Pancost et al., 2011; Dawson et al., 2012; Villanueva et al., 2014), extended archaeol is restricted to alkaliphilic and non-alkaliphilic haloarchaea (e.g., De Rosa, 1982; Teixidor et al., 1993; Dawson et al., 2012) and, in traces, to some methanogens (e.g., Grant et al., 1985; Becker et al., 2016). Archaeol and extended archaeol were also found in various halophilic archaea from recent Lake Magadi (e.g., *Natronobacterium pharaonis*, *Natronobacterium magadii*, *Natronobacterium gregoryi*, *Natronococcus occultus*; Tindall et al., 1985) and haloarchaea are abundant in recent Lake Magadi hot spring communities (Kambura et al., 2016). It is therefore likely that these halophiles have contributed the archaeols to the Lake Magadi cherts.

Cyanobacterial contribution to primary production is directly evidenced by 6Me-$C_{17}$ in LM-1696 (Fig. S1e), which is typically produced by the nitrogen-fixing thermophile *Fischerella* (Coates et al., 2014). Bacterial activity in the chert environment is also indicated by the $C_{32}$-hopanoic acid in LM-1693 and LM-1694, an early degradation product of bacteriohopanepolyols (Farrimond et al., 2002). Further molecular traits of bacteria are $C_{15}$ and $C_{17}$ *i-/ai*-fatty acids (cf.,



Parkes and Taylor, 1983) and monounsaturated and saturated $C_{16}$ and $C_{18}$ fatty acids, although the latter can also derive from algae (e.g., Taipale et. al, 2013, 2016) or higher plant polymers (Kolattukudy, 1980).

The monoethers found in the High Magadi Bed chert bitumens occur in various bacteria, and are particularly prevalent in sulfate reducers (e.g., Yang et al., 2015; Vinçon-Laugier et al., 2016 and references therein). Given the hydrothermally influenced setting, the broad variety of these compounds in the Magadi cherts ($C_{15}$ to $C_{19}$ moieties) may be attributed to thermophiles. Indeed, $i$-$C_{16:0}$, $C_{16:0}$ and $ai$-$C_{17:0}$ monoethers are dominant in Thermodesulfobacteria (Langworthy et al., 1983; Hamilton-Brehm et al., 2013), while $C_{18:1}$ and $C_{18:0}$ monoethers were reported from Aquificales (Huber et al., 1992; Jahnke et al., 2001). The most abundant monoether in the Magadi cherts, 10Me-$C_{16:0}$, was recently detected in mesophilic heterotrophic Desulfobacterales, i.e. sulfate-reducing bacteria (Vinçon-Laugier et al., 2016).

All archaeal lipids in bitumens show an enrichment in $^{13}C$ compared to the fatty acids ($\Delta$ up to +14.6 ‰ between archaeol and short-chain fatty acids in LM-1694). Such heavy values are known from $CO_2$-limited hypersaline environments (e.g., Schidlowski et al., 1984; Schouten et al., 2001) and may be amplified by high bioproductivity (e.g., de Marais et al., 1989; Schidlowski et al., 1994). Halobacteria, however, are heterotrophs and use an organic rather than an inorganic carbon source (e.g., Tindall, 1984; Dawson et al., 2012). If so, these organisms must have fed on an isotopically heavy, thus autochthonous pool of primary produced organic matter (cf., Birgel et al. 2014). This is also in good agreement with the fact that all monoethers are enriched in $^{13}C$ compared to other lipids (Table 4) and underpins that the cherts formed in an evaporitic environment.

### 4.2.2 Eukaryotes

Tetrahymanol is typically produced by ciliates (Mallory et al., 1963; Harvey and McManus et al., 1991), but may also originate from few bacteria (e.g., Kleemann et al., 1990; Banta et al., 2015), ferns (Zander et al., 1969) and fungi (Kemp et al., 1984). It is furthermore associated with alkaline environments (e.g., ten Haven et al., 1989; Thiel et al., 1997), which is well in line with the evaporative setting of Lake Magadi.

The presence of only small amounts of typical algal sterols (cholesterol and sitosterol; cf.,Taipale et al., 2016) in the Lake Magadi cherts indicates minor contributions from these primary producers. Long-chain alkanoic acids and alkan-1-ols with an OEP29 of <<1 in bitumens as well as the corresponding $n$-alkanes with an OEP31 of 0.7–0.9 in kerogens (Table 3) indicate inputs from higher land plants (Eglinton and Hamilton, 1967). Further biomarker evidence for plant input is provided by functionalized sesqui- and diterpenoids (e.g., Otto and Simoneit, 2002; Hautevelle et al., 2006; Fig. 3a). However, the overall predominance of prokaryotic biomarkers (Sec. 4.4.2) and the petrographic observations of silicified microbial mat remains and microbial cells (Sec. 3.1; Fig. 1e–h) suggest a minor importance of eukaryotes in the lake ecosystem.

The $\delta^{13}C$ signals of long-chain $n$-alkanoic acids from Lake Magadi (between −22.4 and −29.6 ‰: Table 4) are in the range of the short-chain homologues ($\Delta$ between +2.4 and −2.6 ‰), while the Icelandic reference sample (−32.5 ‰) shows a pronounced $\delta^{13}C$ depletion ($\Delta$ +7.3 ‰). All $\delta^{13}C$ values are in the range of $C_3$ plants (Schidlowski, 2001). The slightly



heavier $\delta^{13}C$ values of compounds in the Magadi cherts may indicate additional contributions by $C_4$ plants (cf., Chikaraishi et al., 2004; Schidlowski, 2001) which are common in latitudes of the Magadi area (Still et al., 2003). Unlike the $n$-alkanoic acids, long-chain $n$-alkanes in LM-1695–1697 bitumens reveal more depleted $\delta^{13}C$ values around −31 ‰, pointing at a different origin (unknown).

### 4.2.3 Hydrothermal impact on organic matter

In all Lake Magadi cherts a narrow, bell-shaped pattern of $n$-alkanes with a maximum around $n$-$C_{21}$ is dominant in the bitumens (Figs. 3a–e, 5a, S1a–h). This $n$-alkane distribution is also present in the bitumen from the Great Geysir reference sample (IC-1700; Figs. 3f, S1i) and has been frequently reported from other hydrothermal sites (e.g., Simoneit, 1984; Weston and Woolhouse, 1987; Clifton et al., 1990; Simoneit et al., 2009). As the hydrothermal system of the Magadi basin consists of a dilute ground water reservoir, deep brines, and recycled lake brines (Eugster, 1970; Jones et al., 1977), it appears plausible that immature organic compounds from the lake environment have been thermally altered by hydrothermal cycling, resulting, *inter alia*, in a loss of functional groups (cf., McCollom and Seewald, 2003; Hawkes et al., 2016; Rossel et al., 2017). Consequently, the $n$-alkanes from bitumens might represent stable thermal alteration products of originally functionalized compounds, such as linear fatty acids and $n$-alkanols.

Such hydrothermal processes may also yield compounds through the *in-situ* cracking of macromolecular organic matter from the cherts (e.g., alkanes and hopanes, see Sec. 3.2.2). However, temperatures of hydrothermal waters from present springs at Lake Magadi are not higher than 86 °C (Eugster, 1970; Jones et al., 1977). Furthermore, two kerogens from the Green Bed cherts still show relatively low Raman-derived $T_{max}$ values (ca. 40–50 °C; Table 2). *In-situ* maturation near hot springs within the lake may therefore not sufficiently explain the presence of thermally mature organic components in all cherts analyzed.

Alternatively, organic matter from older lake sediments (Oloronga Beds) may have been penetrated by hot fluids, resulting in the formation of hydrothermal petroleum, a process known from other hydrothermal environments (e.g., Clifton et al., 1990; Weston and Woolhouse, 1987; Czochanska et al., 1986; Leif and Simoneit, 1995). This is in good accordance with the early to peak oil window maturity of some bitumen compounds as e.g. indicated by the MPI-1 ratios (≤1.02) and $C_{31}$ S/(S+R) ratios (≤0.61; Peters et al., 2005; Table 2). Cooling and pressure decline of ascending hydrothermal fluids would have led to decreasing solubility of the compounds entrained, resulting in precipitation and thus, fractionation (cf., Simoneit, 1984; Clifton et al., 1990). Such hydrothermal "geochromatography" (Krooss et al., 1991) may explain the narrow distribution of medium-chain $n$-alkanes present in the chert bitumens. Hydrothermal petroleum generation may furthermore be supported by the unimodal distribution patterns of medium-chain $n$-alkan-2-ones and $n$-alkan-2-ols in bitumens, although the exact origin of these compounds is difficult to elucidate. $n$-Alkan-2-ones with similar distributions have previously been reported from hydrothermal oils and may originate from pyrolysis of aliphatic moieties (with $n$-alkan-2-ols as intermediates; Leif and Simoneit, 1995) or pyrolysis of fatty acids with subsequent β-oxidation and decarboxylation (George and Jardin, 1994). Further, both, $n$-alkan-2-ones and $n$-alkan-2-ols were experimentally produced by Fischer-Tropsch-type reactions under



hydrothermal conditions (Rushdi and Simoneit, 1999; Mißbach et al., 2018). In addition to these thermally driven reactions, *n*-alkan-2-ones may also derive from microbial oxidation of *n*-alkanes (e.g., Cranwell, 1987; van Bergen et al., 1998), potentially also with *n*-alkan-2-ol intermediates (Allen et al., 1971; Cranwell et al., 1987).

The relatively low abundance of PAHs in the bitumens may indicate low formation temperatures of hydrothermal petroleum
(cf., Simoneit, 1984; Simoneit et al., 1987; Clifton et al., 1990). This could be due to a shallow sedimentary source which is well in line with the geological situation at Lake Magadi. The Oloronga Beds (maximal thickness of 45 m; Behr, 2002) are the oldest sediments in the young rift basin (ca. 7 Ma; Baker 1958, 1986; geothermal gradient of ca. 200 °C km$^{-1}$; Wheildon et al., 1994) and were not deeply buried at the time of the Green Bed chert deposition. PAHs are common in dissolved organic matter from hydrothermal fluids (e.g., Konn et al., 2009, 2012, McCollom et al., 2015; Rossel et al., 2017), but may
also derive from wildfires. Incomplete combustion of biomass may be a relevant source particularly in LM-1694 and LM-1699, as Flu/(Flu+Py) ratios of about 0.61 (see Table 2) are considered indicative for a wildfire origin (Yunker et al., 2002). Hydrothermal activity may not only have impacted the bitumens. Kerogen from LM-1697 shows highly mature graphitic particles (Raman-based $T_{max}$ of ca. 440°C; Fig. 2; Table 2). These particles may either originate from hydrothermal processes (Luque et al., 2009; van Zuilen et al., 2012), or alternatively from wildfires (Cope and Chaloner, 1980; Schmidt and Noack,
2000, and references therein). As the Flu/(Flu+Py) ratio in the LM-1697 kerogen is substantially lower as expected for a wildfire source (0.23 vs. 0.61; Yunker et al., 2002) and also the bitumen fraction shows no indication of biomass combustion (Flu/(Flu+Py) = 0.79; Table 2), the high temperature particles in LM-1697 most likely do not originate from combustion. We propose that the graphite was produced at depth through the hydrothermally mediated alteration (cf., Luque et al., 2009) of the surrounding trachyte and/or by mineral-templated growth (cf., van Zuilen et al., 2012) during hydrothermal circulation
of bitumen-rich fluids. The hydrothermal fluids may then have transported graphite particles into the lake. Like graphite, thermally altered macromolecular particles from older lake sediments may have also been introduced by hydrothermal fluids which would explain the elevated mean Raman temperatures of LM-1692–1693 and LM-1696 kerogens (Raman-based $T_{max}$ of 100–110°C; Table 2), the high MPI-1 (up to 1.69) and $R_c$ (up to 1.41 %; Table 2) in all kerogens and the strong aromatic character of LM-1692 and 1693 kerogens (see Sec. 3.3).
The occurrence of thermally mature organic components in the studied materials is therefore most likely due to syndepositional hydrothermal processes and reflects an environmental signature.

### 4.3 Organic signatures from the Magadi cherts: implications for the Archean

A fraction of the organic matter preserved in the Magadi cherts may have been introduced by hydrothermal fluids. Such hydrothermally driven redistribution of organic matter has recently been proposed as an important process for kerogen in
Archean hydrothermal vein cherts ("hydrothermal pump hypothesis"; Duda et al., 2018). Another similar feature to findings from Archean cherts is the variability of organic matter characteristics on a small spatial scale observed in the Lake Magadi samples (i.e., heterogeneous thermal maturities of organic matter within a given sample; appearance of organic matter in clots, layers or carbonate rhombs; see Ueno et al., 2004; Allwood et al., 2006; Tice & Lowe, 2006; Glikson et al., 2008;




Morag et al., 2016). In case of ancient cherts, such heterogeneities are generally interpreted as a result of post-depositional metamorphic processes (e.g., Ueno et al., 2004; Tice & Lowe, 2006; Morag et al., 2016) rather than syndepositional hydrothermal activity (e.g., Allwood et al., 2006; Glikson et al., 2008). Our results highlight the possibility that organic matter of very different nature and maturity may be enclosed into hydrothermal chert precipitates *a priori*. Such *in-situ*

mixing of different organic components should also be considered in the interpretation of Archean environments (see Allwood et al., 2006; Glikson et al., 2008; Morag et al., 2016; Duda et al., 2018). Of course, the heterogeneous maturity signals still have to be in accordance with the overall metamorphic history of the host rock.

In addition, our kerogen data show that archaeal lipid biomarkers are preserved in the macromolecular network (Fig. 6). Their presence in the kerogens that show a high thermal overprint (LM-1692–1693; Table 2) implies a rapid incorporation

into macromolecular organic matter, while hydrothermal alteration is active. The kerogen matrix can form an effective shield against oxidation, biodegradation and thermal maturation, thus promoting the preservation of bound compounds over geological time. It has been shown that archaeal lipids can be bound rapidly into macromolecular networks in non-hydrothermal marine sediments (Pancost et al., 2008). The fact that archaeal lipid biomarkers were yielded during the high temperature HyPy step (up to 520 °C) of the Magadi kerogens evidences that these compounds may survive mild diagenetic

influences. In this view, a conservation of kerogen-bound molecular biosignatures also in early Archean hydrothermal cherts (see Marshall et al., 2007; Duda et al., 2018) appears plausible. Our results, together with current findings from Archean hydrothermal systems (Duda et al., 2016, 2018; Djokic et al., 2018), therefore underline the enormous potential of hydrothermal cherts as valuable archives for biosignatures of early life on Earth.

## 5 Conclusions

The depositional record of Lake Magadi (Kenya) contains cherts with different organic matter, remarkably similar to Archean cherts from the Pilbara Craton (Western Australia) and the Barberton Greenstone Belt (South Africa). We found that a significant portion of the bitumens (extractable) and kerogens (non-extractable) in cherts from Lake Magadi is thermally immature and contains biomarkers of various prokaryotic microorganisms (e.g., thermophilic cyanobacteria, sulfate reducers, and haloarchaea). The presence of thermophilic organisms is well in line with a hydrothermal environment.

At the same time, both the bitumens and kerogens also exhibit a thermally mature fraction. We explain this apparent offset between different maturity parameters in the Lake Magadi cherts (immature vs. mature) as a result of a syndepositional hydrothermal alteration (e.g., defunctionalization, pre-maturation) and redistribution of organic matter in the environment. These processes include hydrothermal petroleum expulsion in underlying sedimentary units (Oloronga Beds) and a subsequent introduction of the thermally mature cracking products into the lake. Our findings aid in the interpretation of

heterogeneous organic signatures in Archean rocks, which may reflect original environmental conditions in some cases. In addition, the preservation of archaeal lipid biomarkers in Magadi chert kerogens demonstrates that biomolecules can survive

destructive hydrothermal processes through rapid polymerization and condensation. In this view, a preservation of kerogen-bound molecular biosignatures even in early Archean hydrothermal cherts appears plausible.

**Supplement**

**Author contributions**

MR, VT, WG, JR and JPD designed the study. MR and JR conducted petrographic analyses. MR conducted organic-geochemical analyses and catalytic hydropyrolysis (HyPy). WG and MR performed Raman spectroscopy. CH and MR conducted μ-XRF measurements. MR wrote the manuscript with contributions from all co-authors.

**Competing interests**

The authors declare that they have no conflict of interest.

**Acknowledgements**

We thank G. Arp, W. Dröse, J. Dyckmans, A. Hackmann, D. Hause-Reitner, H. Mißbach, A. Reimer, and B. Röring for scientific and technical support. We furthermore thank A. Schito for providing Raman fitting parameters. This work was financially supported by the International Max Planck Research School (IMPRS) for Solar System Science at the University of Göttingen, the Deutsche Forschungsgemeinschaft (grants DU 1450/3-1 and DU 1450/4-1), and the Göttingen Academy of
Sciences and Humanities.

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





**Figure 1.** Petrographic characteristics of Lake Magadi cherts. (a) Polished slice of LM-1692, revealing organic matter in the silica matrix (arrows). The cross marks the spot for Raman mapping (detailed in b–c). (b) Area of Raman mapping at -2.5 µm (dashed box, image scan 94x72 pixels). (c) Raman mapping result, yellow color indicates high abundances of organic matter. (d) SEM image from LM-1692, showing a porous matrix of microcrystalline quartz. (e) Silicified bacterial filaments from LM-1699 under polarized light (see arrows). (f) Brecciated texture (arrows) in LM-1699. (g) Cloudy microbial features (arrows) in LM-1694. (h) Layered organic matter (arrow) in LM-1693. (i–k) Laminated microbial mat patterns (arrows) preserved in LM-1695 (i), LM-1697 (j) and LM-1698 (k). (l) Silica sinter from Great Geysir, Iceland (IC-1700). (m) Carbonate rhombs (arrows) enclosed in the chert matrix of LM-1696. The box marks the area for µ-XRF scanning (n–q). (n)





Close-up of boxed area showing carbonate rhombs under reflected light. (o–q) µ-XRF analyses of the same area showing silica (o), calcium (p), and sulfur (q) distributions (a brighter color indicates a higher concentration).







**Figure 2.** Raman spectroscopy of kerogen isolated from Green Bed chert sample LM-1697. (a) Kerogen flakes under reflected light. (b) Detail from (a; dashed box) showing selected spots analyzed via Raman (arrows). (c, d) Raman spectra obtained from several spots on the kerogen flakes, including those denoted in (b); insets magnify the spectral range of ca. 1100–1800 rel. cm$^{-1}$ and show fits representative for kerogen populations (band order in c: S, Dl, D, Dr, Gl, G; Rebelo et al., 2016; band order in d: D1, D3, G, D2; Beyssac et al., 2002). Note the close spatial association of kerogen populations of low (immature; c) and high thermal maturity (graphitic; d) within the same sample.







**Figure 3.** Total ion chromatograms (TICs; 10–65 min) of the derivatized bitumens (alcohols were measured as trimethylsilyl ethers, carboxylic acids as methyl esters) from the High Magadi Bed cherts (a–c), the Green Bed cherts (d, e), and the Great Geysir silica sinter (f). Note pronounced $n$-alkanes showing a bell-shaped distribution in the medium-chain range (maxima at $n$-C$_{20}$, $n$-C$_{21}$ or $n$-C$_{22}$) in all chromatograms except LM-1697 (maximum at $n$-C$_{31}$). Other prominent compounds are hexa- and octadecanoic acid (all samples), tetrahymanol (LM-1695, LM-1697, LM-1698), glycerol monoethers (LM-1692–1693), and the glycerol diethers archaeol ($\delta^{13}$C$_{\text{V-PDB}}$ between −18.5 and −22.2 ‰) and extended archaeol ($\delta^{13}$C$_{\text{V-PDB}}$ between −18.3 and −19.9 ‰; LM-1692–1693 and LM-1695). Siloxanes and phthalates were identified as contaminants.



**Figure 4.** Partial GC-MS ion chromatograms (10–60 min) of the derivatized alcohol/ketone (F2; alcohols were measured as trimethylsilyl ethers) and polar (F3; carboxylic acids were measured as methyl esters) fractions from bitumen of the High Magadi Bed chert LM-1694. Alkanoic acids ($m/z$ 74; a) and alkan-1-ols ($m/z$ 75; b) show a clear even-over-odd-predominance and dominances of linear $C_{16}$ and $C_{18}$ homologues. (c) Distribution of glycerol monoethers ($m/z$ 205).



**Figure 5.** Partial GC-MS ion chromatograms (10–55 min) of the hydrocarbon (F1) and derivatized alcohol/ketone fraction (F2; alcohols were measured as trimethylsilyl ethers) from bitumen of the Green Bed chert LM-1698. Medium-chain (~$C_{20}$) alkanes ($m/z$ 85; a), alkan-2-ols ($m/z$ 117; b) and alkan-2-ones ($m/z$ 85; c) show similar distributions with no chain-length-predominance, while long-chain compounds reveal a clear odd-over-even-predominance particularly for alkanes and alkan-2-ols. Hydrothermal cracking of kerogen may produce alkanes that are then converted into alkan-2-ols and subsequently alkan-2-ones (Leif and Simoneit, 1995).





**Figure 6.** Partial GC-MS ion chromatograms (*m/z* 85; 10–70 min) from kerogen HyPy pyrolysates (high temperature step, up to 520 °C) of High Magadi Bed cherts (a–c), and Green Bed cherts (d, e). $\delta^{13}C_{V-PDB}$ values are given for selected compounds. Note different *n*-alkane distributions in the kerogens, with LM-1698 showing the broadest range (*n*-$C_{15}$ to *n*-$C_{46}$). Also note that the regular acyclic isoprenoids phytane and 2,6,10,14,18-pentamethyl icosane are only present in High Magadi Bed chert kerogens (a, b, c). Furthermore, farnesane, norpristane and pristane appear in LM-1695, while biphytane is visible in LM-1693 and LM-1695.



**Table 1.** Geochemical bulk data (C, N, S)

| | $C_{org}$ | | $C_{inorg}$ | | calc. $CaCO_3$ | | N | | S | |
|---|---|---|---|---|---|---|---|---|---|---|
| | wt.% | ± | wt.% | ± | wt.% | ± | wt.% | ± | wt.% | ± |
| LM-1692 | 0.13 | 0.001 | | | | | 0.004 | 0.001 | 0.002 | 0.001 |
| LM-1693 | 0.34 | 0.003 | 0.01 | 0.001 | 0.11 | 0.001 | 0.009 | 0.001 | 0.002 | 0.001 |
| LM-1694 | 0.21 | 0.002 | 0.54 | 0.005 | 4.47 | 0.04 | 0.024 | 0.012 | 0.048 | 0.024 |
| LM-1695 | 0.04 | 0.002 | 0.01 | 0.001 | 0.05 | 0.003 | 0.002 | 0.001 | 0.005 | 0.001 |
| LM-1696 | 0.03 | 0.002 | 0.04 | 0.002 | 0.29 | 0.003 | 0.001 | 0.001 | 0.002 | 0.001 |
| LM-1697 | 0.02 | 0.001 | 0.13 | 0.001 | 1.06 | 0.01 | 0.004 | 0.001 | 0.001 | 0.001 |
| LM-1698 | 0.02 | 0.001 | 0.08 | 0.004 | 0.68 | 0.007 | 0.002 | 0.001 | 0.009 | 0.001 |
| LM-1699 | 0.03 | 0.002 | 0.01 | 0.001 | 0.11 | 0.001 | 0.001 | 0.001 | 0.003 | 0.001 |
| IC-1700 | 0.01 | 0.001 | | | | | 0.004 | 0.001 | 0.003 | 0.001 |

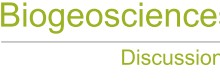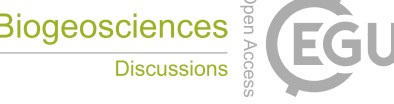


**Table 2.** Environmental and maturity parameters from biomarker analysis (GC-MS) and Raman spectroscopy. The $C_{31}$ hopane S/(S+R) ratios (3rd column) are all in the range of 0.5 to 0.6 and therefore near saturation, implying that most organic compounds may have reached early-oil-window (vitrinite reflectance ≥0.6; Killops and Killops, 2005). These results are fully consistent with reflectances inferred from MPI-1 (5th and 6th column). Raman data (right-most two columns) have been

5 acquired on few specific sample points and therefore reflect the heterogeneity of the sample rather than its bulk properties.

| | Pr/Ph[a] | Ph/$n$-C$_{18}$[b] | $C_{31}$ S/(S+R)[c] | Phe/MP[d] | MPI-1[e] | %R$_c$[f] | Flu/(Flu+Py)[g] | %R$_0$[h] | T$_{max}$ [°C][i] |
|---|---|---|---|---|---|---|---|---|---|
| *Bitumen* | | | | | | | | | |
| LM-1692 | 0.10 | 0.49 | 0.50 | 0.42 | 1.02 | 0.94 | 0.69 | | |
| LM-1693 | 0.18 | 0.36 | 0.58 | 0.62 | 0.63 | 0.66 | 0.66 | | |
| LM-1694 | 0.06 | 0.29 | 0.55 | 0.37 | 0.84 | 0.81 | 0.60 | | |
| LM-1695 | 0.37 | 0.39 | 0.49 | 0.61 | 0.69 | 0.70 | 0.48 | | |
| LM-1696 | 0.21 | 0.31 | 0.59 | 0.71 | 0.56 | 0.61 | 0.74 | | |
| LM-1697 | 0.25 | 0.37 | 0.61 | 0.95 | 0.48 | 0.56 | 0.79 | | |
| LM-1698 | 0.09 | 0.26 | 0.57 | 0.19 | 0.99 | 0.91 | 0.77 | | |
| LM-1699 | | 0.29 | | | | | 0.61 | | |
| IC-1700 | 0.36 | 0.35 | | 0.93 | 0.59 | 0.63 | 0.96 | | |
| *Kerogen* | | | | | | | | | |
| LM-1692 | | 1.89 | | 0.22 | 1.25 | 1.10 | 0.32 | 0.72 | 110 |
| LM-1693 | | 0.49 | | 0.10 | 1.69 | 1.41 | 0.44 | 0.69 | 110 |
| LM-1694 | | | | | | | | 0.54 | 90 |
| LM-1695 | 0.24 | 0.99 | | 0.53 | 1.03 | 0.94 | 0.33 | 0.51 | 80 |
| LM-1696 | | | | | | | | 0.63 | 100 |
| LM-1697 | | | | 0.56 | 1.00 | 0.92 | 0.23 | 0.32 | 40 |
| | | | | | | | | | 440[j] |
| LM-1698 | | | | 0.72 | 0.89 | 0.85 | 0.32 | 0.35 | 50 |

[a]Pristane(Pr)/phytane(Ph) ratio

[b]Phytane(Ph)/$n$-octadecane($n$-C$_{18}$) ratio

[c]17α, 21β(H)-C$_{31}$ hopane 22S/(S+R) ratio

[d]Phenanthrene(Phe)/methylphenanthrene(MP) ratio

10 [e]Methylphenanthrene index = 1.5·(2-MP+3-MP)/(Phe+1-MP+9-MP); Radke and Welte, 1983

[f]Computed vitrinite reflectance = 0.7·MPI-1+0.22 (Boreham et al., 1988), if Phe/MP <1(Brocks et al., 2003a)

[g]Fluoranthene(Flu)/(Flu+pyrene(Py)) ratio



[h]Vitrinite reflectance, calculated from Raman band ratio RA2 (Schito et al., 2017)

[i]Mean maximum temperature, calculated from $R_0$ (Barker and Pavlewicz, 1994)

[j]Mean maximum temperature, calculated from Raman band ratio R2 (Beyssac et al., 2002)





**Table 3.** Odd-to-even-predominances (OEPs; Scalan and Smith, 1970) in bitumens and kerogens

| | Alkanoic acids | | Alkan-1-ols | | Alkan-2-ols | | Alkan-2-ones | | n-Alkanes | |
| | OEP 15 | OEP 29 | OEP 17 | OEP 29 | OEP 21 | OEP 29 | OEP 21 | OEP 29 | OEP 21 | OEP 31 |
|---|---|---|---|---|---|---|---|---|---|---|
| *Bitumen* | | | | | | | | | | |
| LM-1692 | 0.2 | 0.3 | 0.1 | <0.1 | 1.1 | | 2.1 | | 1.2 | 2.8 |
| LM-1693 | 0.4 | 0.4 | 0.2 | 0.1 | 1.0 | 2.3 | 1.5 | | 1.1 | 3.3 |
| LM-1694 | 0.1 | 0.3 | 0.1 | 0.1 | 1.0 | 5.9 | 1.8 | | 1.1 | 5.2 |
| LM-1695 | 0.1 | 0.3 | 0.1 | 0.1 | 1.0 | | 1.3 | | 1.1 | 3.5 |
| LM-1696 | 0.2 | 0.4 | 0.1 | <0.1 | 1.0 | 2.2 | 1.3 | | 1.1 | 2.3 |
| LM-1697 | 0.2 | 0.5 | 0.2 | 0.3 | 0.9 | 1.9 | 1.1 | 1.2 | 1.0 | 6.5 |
| LM-1698 | 0.1 | 0.5 | 0.2 | 0.2 | 1.0 | 1.9 | 1.1 | 1.5 | 1.0 | 7.0 |
| LM-1699 | 0.2 | 0.4 | 0.1 | 0.1 | 0.8 | 4.5 | | | 1.0 | 4.0 |
| IC-1700 | 0.1 | 0.2 | 0.2 | <0.1 | 1.1 | | 1.0 | | 1.1 | 1.8 |
| *Kerogen* | | | | | | | | | | |
| LM-1692 | | | | | | | | | 1.0 | 0.9 |
| LM-1693 | | | | | | | | | 1.0 | 0.7 |
| LM-1695 | | | | | | | | | 1.0 | 0.7 |
| LM-1697 | | | | | | | | | 1.0 | 0.8 |
| LM-1698 | | | | | | | | | 1.0 | 0.7 |

$$OEPn = (C_{n-2}+6 \cdot C_n+C_{n+2})/(4 \cdot C_{n-1}+4 \cdot C_{n+1})^{((-1)^{(n+1)})}$$





**Table 4.** Mean $\delta^{13}C_{\text{V-PDB}}$ values in ‰ of key compound classes and selected biomarkers in bitumens and kerogens

| | LM-1692 | LM-1693 | LM-1694 | LM-1695 | LM-1696 | LM-1697 | LM-1698 | LM-1699 | IC-1700 |
|---|---|---|---|---|---|---|---|---|---|
| *Bitumen* | | | | | | | | | |
| Long-chain *n*-alkanoic acids (C$_{24-28}$) | −25.2 | −27.0 | −26.3 | −28.9 | −22.4 | −25.1 | −27.4 | −29.6 | −32.5 |
| Long-chain *n*-alkan-1-ols (C$_{24-32}$) | −25.0 | −32.3 | −20.2 | −29.1 | −25.1 | −23.7 | −23.2 | −24.0 | −26.1 |
| Long-chain *n*-alkanes (C$_{25-33}$) | | | | −30.9 | −30.1 | −31.5 | −26.2 | −26.5 | |
| Short-chain *n*-alkanoic acids (C$_{12-18}$) | −27.6 | −26.5 | −26.8 | −28.9 | −24.0 | −25.8 | −25.6 | −27.0 | −25.2 |
| Short-chain *n*-alkan-1-ols (C$_{12-18}$) | −33.5 | −32.2 | −35.9 | −30.5 | −32.6 | −33.1 | −29.4 | −31.8 | −27.7 |
| Medium-chain *n*-alkanes (C$_{17-24}$) | −32.1 | −31.7 | −31.7 | −32.8 | −32.6 | −33.3 | −33.3 | −29.7 | −35.7 |
| Phytane | −33.3 | −30.9 | −30.0 | −36.1 | −34.7 | −33.8 | −35.3 | | −38.6 |
| Archaeol | −21.7 | −18.5 | −12.2 | −22.2 | −14.8 | | | −16.6 | |
| Extended archaeol | −18.3 | −19.9 | −15.3 | −19.4 | −19.6 | | | | |
| Monoethers | −20.2 | −20.2 | −10.9 | −18.6 | | | | | |
| *Kerogen* | | | | | | | | | |
| Long-chain *n*-alkanes (C$_{25-40}$) | −27.6 | −30.2 | | −24.9 | | −21.9 | −27.1 | | |
| Medium-chain *n*-alkanes (C$_{17-24}$) | −30.5 | −31.4 | | −28.3 | | −23.5 | −34.2 | | |
| Phytane | −25.1 | −26.8 | | −28.5 | | | | | |
| PMI$_{reg}$ [a] | −22.0 | −24.0 | | −24.6 | | | | | |

[a] 2,6,10,14,18-pentamethylicosane (regular acyclic C$_{25}$ isoprenoid)