# Peer review of "Organic signatures in Pleistocene cherts from Lake Magadi (Kenya)—implications for early Earth hydrothermal deposits"

_Biogeosciences, 2018_

## Referee Comment (RC1) · 5 Feb 2019

jan de Leeuw (Referee)

jan.de.leeuw@nioz.nl

- Title. The title suggests more than the contents of the full article, since: 1) Cyanobacteria, Algae, Higher plants, ciliates, fungi and many bacteria and Archaea present in the Pleistocene setting were not present during the "early Earth", i.e. the (early) Archean 2) most, if not all, hydrothermal vents in the early Archean were at the bottom of the oceans, a setting very different from the Pleistocene setting investigated. The analogy is therefore limited to the syngeneity of immature and mature organic matter as a result of the hydrothermal pump hypothesis. -Extracts and Kerogens. The authors have analysed the extracts as such by GC/MS. High molecular weight compounds

such as Intact GDGTs or their lipid cores, polyesters, etc. (compounds expected to be present in these immature sediments in relative high concentrations), have been missed since the extraction method was not sufficient for extracting such compounds and/or they cannot be analysed by GC/MS. A more polar extraction method in combination with base- and/or acid hydrolysis of extracts and LC/MS analysis would have opened the analytical window very considerably. The kerogens obtained have not been hydrolysed either, so that non-extracted moderate polar, partly high molecular weight, compounds were not removed and analysed by GC/MS or LC/MS. This implies that the HyPy results of the non-hydrolysed kerogens may be biased by pyrolysis products of relatively polar, high molecular weight immature organic matter which is not the result of hydrothermal maturation. It's also possible that the "kerogen" contains high molecular weight compounds produced through sulfurization of immature functionalized low molecular compounds. I understand very well that the authors have limited themselves analytically. That's OK as long as the consequences of such a narrow analytical window is considered in the results and discussion paragraphs. -Stable carbon isotopes. Table 4 . Sample LM-1694 seems to have highly deviating isotope values. Why is that? A HyPy m/z 85 trace might be added to Figs 4 and 6. -UMC. The authors note the presence of a clear UMC in sample LM-1697. UMCs are often the consequence of bacterial biodegradation. In this case it's not clear when this happened, shortly after deposition or recently due to bacterial infection of the outcrop samples. A UMC is also recognized in some of the other samples. I suggest that the authors discuss this topic in more detail.

For future work related to "kerogen" analysis I suggest that the authors consider to apply Thermally assisted hydrolysis (TMH) in combination with GC/MS (see for example K.G.J. Nierop et al., J. of Anal. and Appl. Pyrolysis, 83, pp 227-231 (2008) instead of HyPy. I'm convinced that by applying this TMH method much more info will be obtained due to the release of functionalized compounds also indicating the mode of chemical binding to the macromolecular matrix, since it can be expected that in this particular case the so called kerogen may partly consist of GDGTs and many other

bio(macro)molecules.

---

## Referee Comment (RC2) · Anonymous Referee #2 · 6 Feb 2019

This study describes the co-occurrence of immature "biolipids" with early to peak oil window maturity "geolipids" in a range of chert samples. Overall, the manuscript provides an interesting case study for such a co-occurrence of organic molecules. The described biolipids appear syngenetic to the samples, as some compounds are constrained to specific environments (e.g. archaeol). However, the thermally mature geolipids can occur in a wide range of settings and are quite abundant.

Previous work on hot springs in New Zealand, for example, recorded petroleum seepage as a result geothermal activity. Therefore, the syndepositional hypothesis is valid from paleoenvironmental settings indicating hydrothermal processes (as in this study).

[Figure]

The addition of in-situ Raman evidence for kerogen of a range of different maturities bolsters the validity of the syndepositional hypothesis for the Lake Magadi samples.

Nevertheless, the authors have not provided any convincing evidence that the occurrence of the geolipids are not an artefact of hydrocarbon contamination - the most parsimonious explanation. While the authors used system blanks to track laboratory contaminants, they did not provide any evidence to account for hydrocarbon contaminants already on the rock samples (prior to laboratory analysis). Such contaminants can be introduced even before sampling/handling and storage. The low organic carbon contents (<0.4 wt%) of the samples makes any introduced contaminants even more visible. In recent years, a range of analytical techniques have been established to quantitatively track hydrocarbons from the outer rock surfaces to the interior. It would have been interesting to see what the results of such a study would have been on the cherts from Lake Magadi.

---

## Referee Comment (RC3) · Anonymous Referee #3 · 20 Feb 2019

This is a detailed study of organic matter in cherts from Lake Magadi. These cherts of Pleistocene age are very well preserved, which makes them an interesting analogue for hydrothermally influenced and metamorphosed cherts of Archean age. These ancient silica deposits are ubiquitous in the ancient rock record, and are often found in relation with seafloor hydrothermal activity. The organic fractions in these ancient cherts have been subject to heated debate; do they represent altered remains of ancient life or are they the result of abiologic reactions (such as e.g. Fischer-Tropsch Type synthesis)? As the authors state in the beginning of their paper, in order to interpret this potential archive of ancient life, a sound understanding is necessary of organic matter preservation and alteration in hydrothermal systems. This can be achieved by studying rare

modern analogs of this type of silica deposit. Many of such studies have focused on preserved life in silica sinters that surround hot springs and geysers. The current study, in contrast, focuses on silica deposits in a pure evaporative setting.

The authors used a suite of analytical techniques to characterize the bitumen and kerogen fractions in these rocks, including a detailed description of molecular biomarkers. Several types of prokaryotes could be identified, including thermophilic organisms that typically thrive in hydrothermal settings. Although the majority of the organic fraction is thermally immature, a range of maturities was found that includes highly altered carbonaceous phases. The authors argue that this is evidence for syndepositional hydrothermal alteration and redistribution within the depositional environment. This conclusion has important implications for the interpretation of heterogeneous carbonaceous fractions in Archean cherts. It shows that a range of maturities can occur in the same rock. Also, it shows that certain biomarker molecules can apparently survive hydrothermal circulation, by sequestration into kerogen. Based on this result they hypothesize that preservation of kerogen-bound molecular biosignatures in early Archean hydrothermal cherts may be possible.

Evaluation:

This is a very nice and detailed study of organic matter in recent, relatively unaltered cherts. Indeed, a good case is made for variable maturity as a result of localized hydrothermal circulation. I have some points of criticism (mostly focusing on the interpretation of the Raman spectral analyses), but these are not critical. There are some issues (as described below) that need to be clarified better, and some references to literature on these issues should be made. Overall, this manuscript can be published after only minor revisions.

Comments:

1) A laser power of 1 mW was used during Raman spectroscopy. These kerogen fractions are very immature, with derived temperatures as low as 40 C. For such unaltered,

fragile material, a laser power of 1mW is quite high. Did the authors test if the laser actually affects the kerogen during analysis? For instance causing alteration, or worse, cause combustion?. This should be demonstrated, by a comparison analysis using lower laser power (e.g. 0.1 mW).

2) The very low temperature of alteration (as low as 40C), and the presence of biomarkers for specific groups of prokaryotes, suggests that the Raman spectra of the organic fractions do not only reflect degree of alteration, but also could reflect the type of biologic precursor. For instance, this is suggested by Qu et al. (2015, Astrobiology, 15, 825-841) for carbonaceous fractions found in e.g. the Rhynie chert and the Bitter Springs chert. This should at least be expressed as a possibility, that the Raman-based geothermometer (I don't know if Schito et al., 2017, actually address this issue) is influenced by the type of biomass.

3) The Raman spectra that are presented in Fig.2 are not of high quality. There is a very low signal to noise ratio. The presented peak-fitting protocol, however, is quite sophisticated and requires a high-quality spectrum. It should be explained in detail then, what the uncertainties actually are of fitting these peaks to the range of Raman spectra that were obtained. Also, in general, the calibration of low-temperature Raman-based geothermometers is quite difficult. The geothermometer of Schito et al. (2017) is quite new. There are other, well-known geothermometers, that should also be applied to check if similar temperatures are obtained. The most important ones are the Raman-based determination of H/C-ratio by Ferralis et al. (2016, Carbon, 108, 440-449) and the D1-peak-based geothermometer of Kouketsu et al. (2014, Island Arc, 23, 33-50).

4)In the Discussion, on page 14 line 1-5, it is said that hydrothermal processes can cause syndepositional variation in kerogen maturity. This is not new, and has particularly been suggested for carbonaceous fractions in the hydrothermal feeder part of the 3.5 Ga Apex Chert, Pilbara, Western Australia. In the papers Olcott et al. (2012, Astrobiology, 12, 160-166) and Sforna et al. (2014, GCA, 124, 18-33), it is suggested that variation in kerogen maturity is linked to multiple episodes of hydrothermal fluid

flow. The authors should better describe this process, and refer to these papers.

5) The last part of the discussion, and end of the conclusions, is quite positive about the prospect of finding biomarkers in kerogen in Archean cherts. The authors argue that this is possible because they find good biomarkers in these hydrothermally in-fluenced cherts at Lake Magadi. However, they should mention that most (if not all) cherts of Archean age have experienced greenschist-facies metamorphism, and that they thus have been buried and heated under pressure for millions of years. That's a very different thermal history than the Pleistocene cherts that are studied here. Time is an important factor. Biomarkers are extremely rare in Archean cherts, and the small fractions that have been described are highly controversial. The authors can work that issue out a bit better, and refer to e.g. French et al. (2015, PNAS, 112, 5915-5920) that described these issues. Nevertheless, the authors have proven an important point, that syndepositional hydrothermal circulation would indeed have created a range of matu-rities, and possibly have caused preservation of kerogen-bound biomarker molecules. That such biomarkers could be found in the Archean, however, remains to be seen.

---

## Author Response (AR1)

**GEORG–AUGUST-UNIVERSITY OF GOETTINGEN**

Geoscience Centre

**Department of Geobiology**

*Reinhardt, Manuel*

[Figure]

Univ.Göttingen▪ GZG▪Abt.Geobiologie▪Goldschmidtstr.3▪37077 Göttingen▪Germany

Goldschmidtstr. 3
37077 Göttingen
Germany
Phone:  +49(0)551-39 13756
E-mail: mreinha@gwdg.de
http://www.
geobiologie.uni-goettingen.de

**Associate editor** *Biogeosciences*

Dr. Marcel van der Meer

Göttingen, 04/18/2019

**Submission of the revised manuscript bg-2018-513**

Dear Marcel van der Meer,

Also on behalf of my co-authors I would like to thank you, Jan W. de Leeuw, and the two anonymous reviewers for the thoughtful comments.

As requested, we included the previously suggested corrections into the manuscript (see replies to RC1–3, March 9th). In order to help you to track our changes and modifications, we attached the following documents:

(i) Reply to your comments

(ii) Replies to all reviewer comments (RC1–3 from March 9th)

(iii) Tracked changes version of the manuscript

(iv) Tracked changes version of the supplement

We trust that the revised manuscript will meet the requirements of you and the reviewers.

Yours sincerely,

Manuel Reinhardt

Comment from the editor: "I do think there is one issue you could spend a bit more time on explaining or clarifying. You use the Pleistocene settings as analogs for Archean hydrothermal cherts and I think you need to spend a little bit more time on this to make it more than a way to "sell" your Pleistocene study. The analogy is comparable deposits, hydrothermal cherts, right? I think it is entirely valid to test your approach, methods and types of analysis on a more modern setting to see what works and what doesn't, what type of information you get etc., before actually working on these really old deposits. I am assuming you are actually going to do that? So that sense it works, in your answer to one of the reviewers you actually mention using methods for these samples you would also use for Archean samples. However, there is a few billion years of evolution between the Archean and Pleistocene, so biologically and on a larger environmental scale these settings are not analogues. Not only in the sense that a lot of organisms were not present in the Archean that were present in the Pleistocene, but even the microorganisms present in the Archean continued to evolve for that enormous amount of time. If the settings are similar, you do expect similar functions in both settings, but how that translates in lipids and isotopic compositions? This, together with all kinds of alterations that take place over time for the Archean organic matter, would make it very useful to study the organic matter from the Pleistocene as complete and detailed as possible and not only using techniques you would use for these really old samples. Things might move in and out of analytical windows with time. So I would like to ask you to put a little bit more time in explaining how you see this analogy in your rebuttal and possibly at some key points in the manuscript as well."

Author's response: We consider the Pleistocene hydrothermal chert-generating environment at Lake Magadi more as a (rare) environmental and not so much as an exact biological analog for Archean cherts. So our study is aimed more on the taphonomy (i.e., earliest diagenesis) of organic matter in hydrothermal chert settings. Hydrothermal environments are generally considered unfavorable for the preservation of organic matter, as hot fluid circulation may cause rapid alteration, and destruction of organic compounds. Our data indicate that microbial lipids (in general; not so much specific biomolecules), if produced during the Archean, might well have had a chance to escape syndepositional hydrothermal degradation and be incorporated in the 'final' (solid) chert matrix (notwithstanding that any long-term preservation would still be dependent on the post-depositional metamorphic regime). Further, Archean hydrothermal cherts often show organic matter with different maturity fractions that are commonly explained by post-depositional metamorphic overprint and contamination rather than genuine environmental signals (e.g., Ueno et al., 2004; Tice & Lowe, 2006; Olcott Marshall et al., 2012; Sforna et al., 2014; Morag et al., 2016). We show that such different maturity fractions may also be of primary origin, resulting from syndepositional hydrothermal alteration (and circulation). In this way, we believe that our observations from Lake Magadi can certainly aid in the future analysis and interpretation of organic matter characteristics from Archean hydrothermal cherts.

Comment from the editor: "In one of your comments you mention CO2 limitation as a possible reason for the 13C enriched lipids in LM-1694. Looking at your table 4, not all compounds are extremely enriched in 13C relative to the other samples. The short chain alkanols and alkanoic acids are very similar to the other samples, mainly the monoethers and archaeol and to a lesser extend extended archaeol and even phytane perhaps a little are 13C enriched. The carbon isotopic composition of archaeol seems a bit variable any way. Of course something like CO2 limitation could play a role, it is a bit strange that you see this mainly in this subset of compounds. In this type of setting it would not be surprising to see other CO2 fixation pathways with different carbon isotopic fractionation, variable relative contribution from different pathways could potentially explain the observed differences. Any way something to think about."

Author's response: We agree and included the possibility of different CO2 fixation pathways into the discussion.

References cited in the reply:

Morag, N., Williford, K. H., Kitajima, K., Philippot, P., Van Kranendonk, M. J., Lepot, K., Thomazo, C., and Valley, J. W.: Microstructure-specific carbon isotopic signatures of organic matter from ~3.5 Ga cherts of the Pilbara Craton support a biologic origin, Precambrian Res., 275, 429–449, 10.1016/j.precamres.2016.01.014, 2016.

Olcott Marshall, A., Emry, J. R., and Marshall, C. P.: Multiple Generations of Carbon in the Apex Chert and Implications for Preservation of Microfossils, Astrobiology, 12, 160–166, 10.1089/ast.2011.0729, 2012.

Sforna, M. C., van Zuilen, M. A., and Philippot, P.: Structural characterization by Raman hyperspectral mapping of organic carbon in the 3.46 billion-year-old Apex chert, Western Australia, Geochim. Cosmochim. Acta, 124, 18–33, 10.1016/j.gca.2013.09.031, 2014.

Tice, M. M., and Lowe, D. R.: The origin of carbonaceous matter in pre-3.0 Ga greenstone terrains: A review and new evidence from the 3.42 Ga Buck Reef Chert, Earth Sci. Rev., 76, 259–300, 10.1016/j.earscirev.2006.03.003, 2006.

Ueno, Y., Yoshioka, H., Maruyama, S., and Isozaki, Y.: Carbon isotopes and petrography of kerogens in ~3.5-Ga hydrothermal silica dikes in the North Pole area, Western Australia, Geochim. Cosmochim. Acta, 68, 573–589, 10.1016/S0016-7037(03)00462-9, 2004.

Biogeosciences Discuss.,
https://doi.org/10.5194/bg-2018-513-AC1, 2019

[Figure]

Comment from referee: "The title suggests more than the contents of the full article, since: 1) Cyanobacteria, Algae, Higher plants, ciliates, fungi and many bacteria and Archaea present in the Pleistocene setting were not present during the "early Earth", i.e. the (early) Archean...,

Author's response: We agree in that the organic matter sources of modern and Archean organic matter were certainly not identical. We therefore will use a more cautious wording of the title, avoiding the term 'analog' in respect to organic matter.

[Figure]

Planned changes in manuscript: The title will be reworded.

. . .2) most, if not all, hydrothermal vents in the early Archean were at the bottom of the oceans, a setting very different from the Pleistocene setting investigated. The analogy is therefore limited to the syngeneity of immature and mature organic matter as a result of the hydrothermal pump hypothesis".

Author's response: We do not agree with the referee here. Recent works demonstrated that a variety of early Archean facies in Barberton and Pilbara areas reflect shallow marine hydrothermal (Allwood et al., 2006; Hickman-Lewis et al., 2018) or even terrestrial hot spring environments (Djokic et al., 2017), and several authors have pointed at the similarities of the Magadi cherts and Archean cherts with respect to their formation and lithology (Brenna, 2016; Eugster and Jones, 1968).

Comment from referee: "The authors have analysed the extracts as such by GC/MS. High molecular weight compounds such as Intact GDGTs or their lipid cores, polyesters, etc. (compounds expected to be present in these immature sediments in relative high concentrations), have been missed since the extraction method was not sufficient for extracting such compounds and/or they cannot be analysed by GC/MS. A more polar extraction method in combination with base- and/or acid hydrolysis of extracts and LC/MS analysis would have opened the analytical window very considerably".

Author's response: We agree that the use of further analytical techniques may have led to the identification of additional compounds. However, our methodology was not designed as being comprehensive with respect to all (polar) compounds present. As we regard our Pleistocene (10–100 ka old) thermally altered samples essentially as 'fossil', we focused more on the GC-amenable alkyl moieties and adhered as much as possible to the techniques typically used on ancient (incl. Archaean) organic matter, i.e., Raman, GC–MS, HyPy, and microscopy. It should nonetheless be noted we did use acidic hydrolysis on the TOEs to cleave ester-bound compounds (TMCS/MeOH), so

that GC-amenable ester-bound moieties have been covered in the bitumen fractions.

Planned changes in manuscript: We will clarify that high molecular weight compounds like GDGTs and polymers were not analyzed in this study (introduction; chapter 3.2.1) and limit our conclusions on archaeal lipid preservation in the Magadi chert kerogens to low/medium molecular weight compounds.

Comment from referee: "The kerogens obtained have not been hydrolysed either, so that non-extracted moderate polar, partly high molecular weight, compounds were not removed and analysed by GC/MS or LC/MS. This implies that the HyPy results of the non-hydrolysed kerogens may be biased by pyrolysis products of relatively polar, high molecular weight immature organic matter which is not the result of hydrothermal maturation".

Author's response:

(i) As outlined above we intentionally used the techniques generally applied to study ancient organic matter, hence for kerogen preparation we strictly adhered to the established and commonly used method of Durand (1980), i.e., sequential extraction with different solvents, HCl-treatment, HF-treatment, and repeated extractions of the resulting residue.

(ii) A potential bias by non-extracted low-molecular weight moieties in the kerogen pyrolysis products ('bitumen II') can be excluded, because these compounds were removed during the pre-heating step (330 °C) in the HyPy runs. Notably, this step yielded only minor products (thus excluded from discussion).

(iii) DCM/MeOH-based solvent combinations similar to those used in our study were reported to successfully extract lipids of higher molecular weight such as GDGTs (e.g., Pancost et al., 2008). We feel that most lipid-like material including GDGT lipids should have been removed from the kerogen during the sequential extraction and acid dissolution steps. However, the possibility that some biomarkers may partly originate from

residual compounds trapped in the 'kerogen' and missed by the extensive kerogen extraction procedure can hardly be fully excluded (and could hardly be ever excluded in kerogen studies).

Planned changes in manuscript: We will mention the possibility that biphytane may partly derive from incomplete removal of GDGTs during extraction and tone down the implications on archaeal lipid preservation in the kerogens accordingly (see author's response to the previous referee comment).

Comment from referee: "It's also possible that the "kerogen" contains high molecular weight compounds produced through sulfurization of immature functionalized low molecular compounds. I understand very well that the authors have limited themselves analytically. That's OK as long as the consequences of such a narrow analytical window is considered in the results and discussion paragraphs".

Author's response: We agree that the sulfurization of immature compounds may contribute, with or without hydrothermal influence, to the formation of early kerogen-like macromolecules. However, the Magadi cherts have extremely low sulfur contents (Table 1) and these tiny amounts of sulfur are mostly hosted in inorganic oxidized species, mostly gypsum. Therefore we do not consider sulfurization as an important process in this setting. Yet the biomarker inventory revealed evidence for lipid inputs from sulfate reducers, and it may be possible that sulfurization could occur in microscale environments where sulfate reduction was active. Unfortunately this possibility could not be further explored using our analytical setup.

Planned changes in manuscript: The occurrence of sulfates will be specified and the possibility of sulfurization will be discussed.

Comment from referee: "Table 4. Sample LM-1694 seems to have highly deviating isotope values. Why is that? A HyPy m/z 85 trace might be added to Figs 4 and 6".

Author's response: Magadi is an evaporitic environment showing varying salinities,

so the 13C enrichment in compounds from LM-1694 may be explained by a higher salinity/evaporation (stronger CO2-limitation) during deposition of that specific chert.

Planned changes in manuscript: As suggested a m/z 85 HyPy trace of LM-1694 will be added to Fig. 6 and the corresponding bitumen TIC to Fig. 3.

Comment from referee: "The authors note the presence of a clear UMC in sample LM-1697. UMCs are often the consequence of bacterial biodegradation. In this case it's not clear when this happened, shortly after deposition or recently due to bacterial infection of the outcrop samples. A UMC is also recognized in some of the other samples. I suggest that the authors discuss this topic in more detail".

Author's response and planned changes in manuscript: We agree and will discuss biodegradation in the manuscript. However, UCMs may also appear in non-biodegraded low-maturity oils (Peters et al., 2005, p. 106) and the Ph/n-C18 value of LM-1697 is not elevated as compared to other Magadi chert bitumens without pronounced UCMs (see Table 2).

Comment from referee: "For future work related to "kerogen" analysis I suggest that the authors consider to apply Thermally assisted hydrolysis (TMH) in combination with GC/MS (see for example K.G.J. Nierop et al., J. of Anal. and Appl. Pyrolysis, 83, pp 227-231 (2008) instead of HyPy. I'm convinced that by applying this TMH method much more info will be obtained due to the release of functionalized compounds also indicating the mode of chemical binding to the macromolecular matrix, since it can be expected that in this particular case the so called kerogen may partly consist of GDGTs and many other bio(macro)molecules".

Author's response: We thank the referee for his suggestion. While we are still convinced that our analytical setup appropriately supports the conclusions drawn in this manuscript, we will certainly consider the TMH method for the design of our future experiments!

[Figure]

References cited in the reply:

Allwood, A. C., Walter, M. R., and Marshall, C. P.: Raman spectroscopy reveals thermal palaeoenvironments of c.3.5 billion-year-old organic matter, Vib. Spectrosc., 41, 190–197, 10.1016/j.vibspec.2006.02.006, 2006.

Brenna, B. L.: The Chemical, Physical, and Microbial Origins of Pleistocene Cherts at Lake Magadi, Kenya Rift Valley, M.Sc. thesis, Department of Geological Sciences, University of Saskatchewan, Saskatoon, 158 pp., 2016.

Durand, B.: Sedimentary organic matter and kerogen. Definition and quantitative importance of kerogen, in: Kerogen: Insoluble Organic Matter from Sedimentary Rocks, edited by: Durand, B., Editions Technip., Paris, 13–34, 1980.

Djokic, T., Van Kranendonk, M. J., Campbell, K. A., Walter, M. R., and Ward, C. R.: Earliest signs of life on land preserved in ca. 3.5 Ga hot spring deposits, Nat. Commun., 8, e15263, 10.1038/ncomms15263, 2017.

Eugster, H. P., and Jones, B. F.: Gels Composed of Sodium-Aluminium Silicate, Lake Magadi, Kenya, Science, 161, 160–163, 10.1126/science.161.3837.160, 1968.

Hickman-Lewis, K., Cavalazzi, B., Foucher, F., and Westall, F.: Most ancient evidence for life in the Barberton greenstone belt: Microbial mats and biofabrics of the ∼3.47 Ga Middle Marker horizon, Precambrian Res., 312, 45–67, 10.1016/j.precamres.2018.04.007, 2018.

Pancost, R. D., Coleman, J. M., Love, G. D., Chatzi, A., Bouloubassi, I., and Snape, C. E.: Kerogen-bound glycerol dialkyl tetraether lipids released by hydropyrolysis of marine sediments: A bias against incorporation of sedimentary organisms?, Org. Geochem., 39, 1359–1371, 10.1016/j.orggeochem.2008.05.002, 2008.

Peters, K. E., Walters, C. C., and Moldowan, J. M.: The Biomarker Guide: I. Biomarkers and Isotopes in the Environment and Human History, 2nd ed., Cambridge University Press., Cambridge, 471 pp., 2005.

[Figure]

[Figure]

Biogeosciences Discuss.,
https://doi.org/10.5194/bg-2018-513-AC2, 2019

[Figure]

Comment from referee: This study describes the co-occurrence of immature "biolipids" with early to peak oil window maturity "geolipids" in a range of chert samples. Overall, the manuscript provides an interesting case study for such a co-occurrence of organic molecules. The described biolipids appear syngenetic to the samples, as some compounds are constrained to specific environments (e.g. archaeol). However, the thermally mature geolipids can occur in a wide range of settings and are quite abundant. Previous work on hot springs in New Zealand, for example, recorded petroleum seepage as a result geothermal activity. Therefore, the syndepositional hypothesis is valid

from paleoenvironmental settings indicating hydrothermal processes (as in this study). The addition of in-situ Raman evidence for kerogen of a range of different maturities bolsters the validity of the syndepositional hypothesis for the Lake Magadi samples.

Comment from referee: Nevertheless, the authors have not provided any convincing evidence that the occurrence of the geolipids are not an artefact of hydrocarbon contamination - the most parsimonious explanation. While the authors used system blanks to track laboratory contaminants, they did not provide any evidence to account for hydrocarbon contaminants already on the rock samples (prior to laboratory analysis). Such contaminants can be introduced even before sampling/handling and storage. The low organic carbon contents (<0.4 wt%) of the samples makes any introduced contaminants even more visible. In recent years, a range of analytical techniques have been established to quantitatively track hydrocarbons from the outer rock surfaces to the interior. It would have been interesting to see what the results of such a study would have been on the cherts from Lake Magadi.

Author's response and planned changes in manuscript: We are aware of this problem and therefore conducted interior vs. exterior experiments on two Magadi cherts. The results support syngeneity of the geolipids and will be included into the supplement.

———————————————————

[Figure]

Biogeosciences Discuss.,
https://doi.org/10.5194/bg-2018-513-AC3, 2019

[Figure]

This is a very nice and detailed study of organic matter in recent, relatively unaltered cherts. Indeed, a good case is made for variable maturity as a result of localized hydrothermal circulation. I have some points of criticism (mostly focusing on the interpretation of the Raman spectral analyses), but these are not critical. There are some issues (as described below) that need to be clarified better, and some references to literature on these issues should be made. Overall, this manuscript can be published after only minor revisions.

Comment from referee: 1) A laser power of 1 mW was used during Raman spec-

troscopy. These kerogen fractions are very immature, with derived temperatures as low as 40 C. For such unaltered, fragile material, a laser power of 1mW is quite high. Did the authors test if the laser actually affects the kerogen during analysis? For instance causing alteration, or worse, cause combustion?. This should be demonstrated, by a comparison analysis using lower laser power (e.g. 0.1 mW).

Author's response: We agree, and we are fully aware of this problem. In our study, laser energy and exposure time were optimized on representative organic-bearing test spots prior to analyzing the actual spots selected for presentation in the manuscript. With the resulting protocol the degradation of organics (during laser irradiation) was found to be minor.

Changes planned: We will describe the laser power test in the "Materials and Methods" section (2.6 Raman Spectroscopy).

Comment from referee: 2) The very low temperature of alteration (as low as 40C), and the presence of biomarkers for specific groups of prokaryotes, suggests that the Raman spectra of the organic fractions do not only reflect degree of alteration, but also could reflect the type of biologic precursor. For instance, this is suggested by Qu et al. (2015, Astrobiology, 15, 825-841) for carbonaceous fractions found in e.g. the Rhynie chert and the Bitter Springs chert. This should at least be expressed as a possibility, that the Raman-based geothermometer (I don't know if Schito et al., 2017, actually address this issue) is influenced by the type of biomass.

Author's response: We agree, this is certainly an important point.

Changes planned: We will refer to the study by Qu et al. (2015) and include the information that the obtained low temperature Raman data possibly reflect both, thermal maturity and the specific type of biological precursor.

Comment from referee: 3) The Raman spectra that are presented in Fig.2 are not of high quality. There is a very low signal to noise ratio. The presented peak-fitting

protocol, however, is quite sophisticated and requires a high-quality spectrum. It should be explained in detail then, what the uncertainties actually are of fitting these peaks to the range of Raman spectra that were obtained. Also, in general, the calibration of low-temperature Ramanbased geothermometers is quite difficult. The geothermometer of Schito et al. (2017) is quite new. There are other, well-known geothermometers, that should also be applied to check if similar temperatures are obtained. The most important ones are the Ramanbased determination of H/C-ratio by Ferralis et al. (2016, Carbon, 108, 440-449) and the D1-peak-based geothermometer of Kouketsu et al. (2014, Island Arc, 23, 33-50).

Author's response: Most Raman geothermometers, including those mentioned in this referee comment, focus on temperatures above 150°C, so we feel that they cannot be usefully applied here. Schito et al. (2017) appear to be the only authors attempting Raman thermometry below 100 °C.

Changes planned: CoD-values (R2) for the fittings and a word of caution (see point 2 above) will be added to the manuscript. The signal-to-noise ratio has been addressed under point 1 (see above).

Comment from referee: 4) In the Discussion, on page 14 line 1-5, it is said that hydrothermal processes can cause syndepositional variation in kerogen maturity. This is not new, and has particularly been suggested for carbonaceous fractions in the hydrothermal feeder part of the 3.5 Ga Apex Chert, Pilbara, Western Australia. In the papers Olcott et al. (2012, Astrobiology, 12, 160-166) and Sforna et al. (2014, GCA, 124, 18-33), it is suggested that variation in kerogen maturity is linked to multiple episodes of hydrothermal fluid flow. The authors should better describe this process, and refer to these papers.

Author's response: We agree.

Changes planned: We will rephrase the respective part (p. 14) and include these papers into our discussion.

[Figure]

Comment from referee: 5) The last part of the discussion, and end of the conclusions, is quite positive about the prospect of finding biomarkers in kerogen in Archean cherts. The authors argue that this is possible because they find good biomarkers in these hydrothermally influenced cherts at Lake Magadi. However, they should mention that most (if not all) cherts of Archean age have experienced greenschist-facies metamorphism, and that they thus have been buried and heated under pressure for millions of years. That's a very different thermal history than the Pleistocene cherts that are studied here. Time is an important factor. Biomarkers are extremely rare in Archean cherts, and the small fractions that have been described are highly controversial. The authors can work that issue out a bit better, and refer to e.g. French et al. (2015, PNAS, 112, 5915-5920) that described these issues. Nevertheless, the authors have proven an important point, that syndepositional hydrothermal circulation would indeed have created a range of maturities, and possibly have caused preservation of kerogen-bound biomarker molecules. That such biomarkers could be found in the Archean, however, remains to be seen.

Author's response: We agree. The post-depositional thermal history of the Magadi cherts is not comparable with Archean hydrothermal deposits. Nevertheless, our data indicate that not all molecular fingerprints, such as lipid biomarkers, are lost during initial hydrothermal heating and mild diagenesis in hydrothermal environments.

Changes planned: We will tone down our positive view and implement the work by French and colleagues.

References cited in the reply:

[revised manuscript text omitted]
$_{24\text{–}28}$) | −25.2 | −27.0 | −26.3 | −28.9 | −22.4 | −25.1 | −27.4 | −29.6 | −32.5 |
| Long-chain *n*-alkan-1-ols (C$_{24\text{–}32}$) | −25.0 | −32.3 | −20.2 | −29.1 | −25.1 | −23.7 | −23.2 | −24.0 | −26.1 |
| Long-chain *n*-alkanes (C$_{25\text{–}33}$) | | | | −30.9 | −30.1 | −31.5 | −26.2 | −26.5 | |
| Short-chain *n*-alkanoic acids (C$_{12\text{–}18}$) | −27.6 | −26.5 | −26.8 | −28.9 | −24.0 | −25.8 | −25.6 | −27.0 | −25.2 |
| Short-chain *n*-alkan-1-ols (C$_{12\text{–}18}$) | −33.5 | −32.2 | −35.9 | −30.5 | −32.6 | −33.1 | −29.4 | −31.8 | −27.7 |
| Medium-chain *n*-alkanes (C$_{17\text{–}24}$) | −32.1 | −31.7 | −31.7 | −32.8 | −32.6 | −33.3 | −33.3 | −29.7 | −35.7 |
| Phytane | −33.3 | −30.9 | −30.0 | −36.1 | −34.7 | −33.8 | −35.3 | | −38.6 |
| Archaeol | −21.7 | −18.5 | −12.2 | −22.2 | −14.8 | | | −16.6 | |
| Extended archaeol | −18.3 | −19.9 | −15.3 | −19.4 | −19.6 | | | | |
| Monoethers | −20.2 | −20.2 | −10.9 | −18.6 | | | | | |
| *Kerogen* | | | | | | | | | |
| Long-chain *n*-alkanes (C$_{25\text{–}40}$) | −27.6 | −30.2 | −22.7 | −24.9 | | −21.9 | −27.1 | | |
| Medium-chain *n*-alkanes (C$_{17\text{–}24}$) | −30.5 | −31.4 | −27.3 | −28.3 | | −23.5 | −34.2 | | |
| Phytane | −25.1 | −26.8 | | −28.5 | | | | | |
| PMI$_{reg}$[a] | −22.0 | −24.0 | −14.5 | −24.6 | | | | | |

[a]2,6,10,14,18-pentamethylicosane (regular acyclic C$_{25}$ isoprenoid)

[Figure]

**Supplementary information for:**

[revised manuscript text omitted]